# Drug resistance and its risk factors among extrapulmonary tuberculosis in Ethiopia: A systematic review and meta-analysis

Getu Diriba[1]*, Habteyes Hailu Tola[1], Ayinalem Alemu[1], Bazezew Yenew[1], Dinka Fikadu Gamtesa[1], Abebaw Kebede[1,2]

1 Ethiopia National Tuberculosis Reference Laboratory, Ethiopian Public Health Institute, Addis Ababa, Ethiopia, 2 Department of Microbial, Cellular and Molecular Biology, College of Natural and Computational Sciences, Addis Ababa University, Addis Ababa, Ethiopia

* getud2020@gmail.com

## Abstract

### Background

Drug-resistant tuberculosis and extrapulmonary tuberculosis are the world major public health issues. Although some primary studies have been reported on the burden of drug-resistant tuberculosis in extrapulmonary tuberculosis patients in Ethiopia, there is no systematic review and meta-analysis that attempt to summarize the available literature. Thus, we aimed to estimates the prevalence of drug-resistance in extrapulmonary tuberculosis patients and summarize the risk factors associated with the occurrence of extrapulmonary tuberculosis in Ethiopia.

### Methods

We conducted a systematic review of the published primary studies on extrapulmonary drug-resistant tuberculosis in Ethiopia.

### Results

Eight observational studies were included in this review from different regions of Ethiopia. The overall pooled prevalence of rifampicin resistance was 6% (95% CI 0.03–0.10), while isoniazid resistance was 7% (95% CI 0.03–0.12). The pooled prevalence of multidrug-resistant tuberculosis was 4% (95% CI 0.01–0.07). Previous tuberculosis treatment history and male gender are frequently reported risk factors for developing drug-resistant tuberculosis in extrapulmonary tuberculosis patients.

### Conclusion

The current review has identified a high proportion of resistance to rifampicin, isoniazid, and multidrug-resistant tuberculosis in patients with extrapulmonary tuberculosis in Ethiopia. Clinicians should request drug susceptibility testing for all patients with presumptive extrapulmonary tuberculosis to detect drug-resistance.

**Data Availability Statement:** All relevant data are within the manuscript.

**Funding:** The author(s) received no specific funding for this work.

**Competing interests:** The authors declare that they have no competing interests.

**Abbreviations:** AFB, Acid-fast bacilli; CI, Confidence interval; DR-TB, Drug-resistant tuberculosis; DST, Drug susceptibility test; EPTB, Extra pulmonary tuberculosis; FNAC, Fine needle aspiration; INH, Isoniazid; MDRTB, Multi-drug-resistant tuberculosis; MGIT, Mycobacterium Growth Indicator Tube; PRISMA, Preferred Reporting Items for Systematic Review and Meta-Analysis; PTB, pulmonary TB; RIF, Rifampicin; RR-TB, Rifampicin resistance Tuberculosis; TB, Tuberculosis; WHO, World Health Organization.

# Introduction

Tuberculosis (TB) is one of the leading causes of death from a single infectious agent [1]. TB mainly affects the lungs (pulmonary TB), but it can also affect other body sites, which refer, to extrapulmonary TB (EPTB) [1, 2]. EPTB is defined as any bacteriologically confirmed or clinically diagnosed case of TB involving organs other than the lungs. The body organs other than the lungs that are mainly affected by TB include the pleura, lymph nodes, abdomen, genitourinary tract, skin, joints, bones, and meninges [3–5]. The incidence of EPTB involvement of TB occurs in approximately 10 million new TB cases that were reported to the World Health Organization (WHO) in 2018, 16% were EPTB cases; incidence rates ranged from 8% in the Western Pacific Region to 24% in the Eastern Mediterranean Region [6].

Drug-resistant (DR) TB is the main challenge of the global TB control program due to its high risk of relapse, treatment failure, prolonged transmission of the bacilli, and death [7]. A recent global estimate indicates about half a million rifampicin-resistant tuberculosis (RR-TB) cases well be developed across the world in 2019 [8]. Of the total RR-TB cases developed in 2019 across the world, 82% had multidrug-resistant tuberculosis (MDR-TB) [2]. MDR-TB is defined as *Mycobacterium tuberculosis* that is resistant to at least isoniazid and rifampicin [9, 10]. Extensively drug-resistant TB (XDR-TB) is also the current challenge that faces the global TB control program. XDR-TB is referred to as *Mycobacterium tuberculosis*. It is resistant to at least isoniazid (INH) and rifampicin (RIF), but also resistant to any of the fluoroquinolones and at least one of the injectable second-line drugs (amikacin, capreomycin, or kanamycin) [11–13]. A total of 8,014 XDR-TB cases were reported from 72 countries in 2017. Combining their data, the average proportion of MDR-TB cases with XDR-TB was 6.2% in 2017 [14].

The treatment of DR-TB is more difficult than the treatment of drug-susceptible TB because it requires the use of second-line drugs that are of a longer duration, more costly, more toxic, and less effective. Patients who are infected with strains resistant to isoniazid and rifampicin are practically incurable by standard first-line TB drugs [13, 15].

Currently, Ethiopia is ranked third in Africa in TB incidence and among the 30 high TB, TB/HIV, and MDR-TB burden countries [10]. The estimated prevalence rate of MDR-TB in Ethiopia is 0.7% among new cases and 16% among previously treated patients in 2019 [1, 10, 16]. Although the burden of MDR-TB is decreasing in the country over time in new cases, it is still high among previously treated cases [1, 2]. For example, the estimated prevalence of MDR-TB in previously treated cases was 17.8% in 2017, while in 2018 it was 14% [2, 17]. Moreover, although several review studies tried to pool the prevalence of DR-TB in pulmonary TB cases, no review study tried to pool the burden of DR-TB in EPTB patients in Ethiopia. Besides, there is no nationwide drug resistance surveillance like pulmonary TB in Ethiopia and the challenges the country is planning to face are not as simple as conducting TB DRS. Therefore, we aimed to estimate the pooled prevalence of DR-TB in patients with EPTB, and summarize the risk factors associated with the occurrence of EPTB in Ethiopia.

# Methods

## Protocol and registration

The protocol of this systematic review and meta-analysis was registered on the PROSPERO (International Prospective Register of Systematic Reviews), University of York. It was assigned a registration number (CRD42020139028) and can be accessed from the link https://www.crd.york.ac.uk/prospero/#record. However, ethical clearance was not sought, because this study was based on already published primary studies.

## Search strategy

We conducted a systematic review and meta-analysis to estimate the pooled prevalence of DR-TB in patients with EPTB in Ethiopia, and to summarize risk factors associated with DR-TB occurrence following the Preferred Reporting Items for Systematic Review and Meta-Analysis (PRISMA) statement guideline for reporting of systematic review and meta-analysis [18, 19]. We systematically searched electronic databases such as MEDLINE (PubMed), ScienceDirect, and Google Scholar from January 15 to 21 February 2021 to retrieve articles published in English without limiting the publication year. We used the keywords: "prevalence"; "burden"; "anti-TB drug susceptibility"; "anti-TB drug resistance"; "Resistant TB"; "Multidrug resistance TB"; "MDR-TB"; "extensively drug resistance" "XDR-TB" and "extrapulmonary tuberculosis" with free text and Medical Subject Headings (MeSH) to retrieve the potential primary studies. We also searched for bibliographies of other reviews and citations of the original articles included in this review.

## Inclusion and exclusion criteria

In the study, cross-sectional and retrospective observational studies were included. The study included studies that mentioned DR-TB prevalence in new and/or previously treated patients with extrapulmonary tuberculosis who were diagnosed using the conventional method of drug sensitivity testing. Studies with inaccurate data, studies not disclosing the DST method, and review articles, meta-analyses, duplicates, and pulmonary TB were all excluded from the study.

## Study selection

Two authors (GD and AA) were searched the electronic databases separately to retrieve the potential studies. All retrieved records from the systematic search were screened using titles and abstracts. Irrelevant records were excluded by the study population and outcome difference. In the next step, full-text articles were screened by two reviewers (GD and DF) independently to select relevant articles based on inclusion criteria. Discrepancies between two reviewers (GD and DF) were addressed by a discussion between the two authors.

## Data extraction

We extracted the required data from each included study using a format prepared for a Microsoft Excel spreadsheet. For each selected study, first author, publication year, sample size, study design, study area, drug susceptibility test (DST) method, number of patients with DST results, INH susceptibility status, RIF susceptibility status, MDR-TB, and risk factors that are associated with drug-resistant TB were extracted. Two authors (AA and DF) extracted the data independently. Discrepancies in the data extraction between the two authors were resolved through discussions.

## Study quality and risk of bias assessment

The Newcastle-Ottawa quality assessment scale was employed to assess the quality of the included studies by two authors (GD and DF) independently. The disagreement between the two authors was resolved by consensus. In the case of a persistent disagreement, a third author was consulted.

## Statistical analysis and data synthesis

We estimated the pooled prevalence of DR-TB in EPTB patients with its 95% confidence interval by a random-effects meta-analysis model, assuming the true effect size varies between the

included studies. The 'metaprop' command on STATA 14 (STATA Corporation, College Station, TX, USA) was used to estimate the pooled prevalence of DR-TB in patients with EPTB. A forest plot was used to display pooled DR-TB prevalence. We assessed heterogeneity in the reported prevalence by chi-square (Q), p-values, and $I^2$ [20]. $I^2 \geq 50\%$ was considered to be the presence of heterogeneity [21]. We also assessed the presence of publication bias with a funnel plot and Egger's test (p-value < 0.1 as significant level).

## Results

### Study selection

The three electronic databases yielded a total of 477 articles, which were then imported into an Endnote library. Fig 1 depicts the article search and inclusion process. We retrieved a total of 477 study records during the search process. A total of 195 studies remained after 280 records were removed due to duplication, population, and outcome differences. In the next step, 136 studies were excluded due to the absence of data on the outcome variable, and 59 studies have remained. Finally, a total of 8 articles met the inclusion criteria after 51 studies were excluded because they were conducted on foreigners and for pulmonary tuberculosis only. Over all, full screening was done based on the preferred reporting items for systematic reviews.

### Characteristics of included studies

On a total of 776 confirmed EPTB patients, eight included studies reported DR (including RIF resistance, INH resistance, and MDR-TB). Of the eight studies, two were reported on RIF resistance only [22, 23], three on RIF and INH resistance [24–26], and another three on all first-line anti-TB drugs [27–29]. Regarding DST methods, three studies used phenotypic methods (proportion, absolute concentration, and the BACTEC system) [27–29], two used the genotypic Xpert MTB/RIF assay [22, 23], and three used the genotypic MTBDR*plus* assay [24–26]. The publication years of the included studies ranged from 2014 to 2020 and all studies employed a cross-sectional study design (Table 1).

Data on a total of 776 patients with EPTB was pooled from eight studies that were included in this review. Of the total of 776 patients, 51 had RIF resistance to bacilli. The overall pooled prevalence of RIF resistance was 6% (95% CI 0.03–0.10, $I^2 = 74.66\%$). The highest RIF resistance proportion was reported from Addis Ababa (22%) [28] and the minimum from north Ethiopia (2%) [26, 27]. There was significant heterogeneity in the reported prevalence rate of RIF resistance (Fig 2).

Of the eight studies included in this review, five reported INH resistance proportion. The crude prevalence of INH resistance has varied across the studies reported from different geographical locations of Ethiopia. Of the included studies, three studies [25, 28, 29] reported a higher prevalence of INH resistance, which ranged from 8% to 15%. The overall pooled prevalence of INH resistance was 7% (95% CI 0.03–0.12, $I^2 = 73.80\%$, p-value test for heterogeneity < 0.01) (Fig 3).

Of the eight studies included in this review, five studies have reported the prevalence of MDR-TB (Table 1). Two studies [25, 29] reported from Addis Ababa and show the highest (8% and 9%) prevalence of MDR-TB. Moreover, two studies [26, 27] reported from northern Ethiopia have shown a low (1%) prevalence of MDR-TB. The overall pooled prevalence of MDR-TB was 4% (95% CI 0.01–0.07, $I^2 = 71.84\%$, p-value < 0.01) (Fig 4).

The presence of publication bias was assessed using funnel plot and Egger statistical tests at a 5% level of significance. The publication bias based on the funnel plot and Egger test results (p = 0.01) also confirmed the presence of publication bias in the included studies in the estimated prevalence of drug resistance.

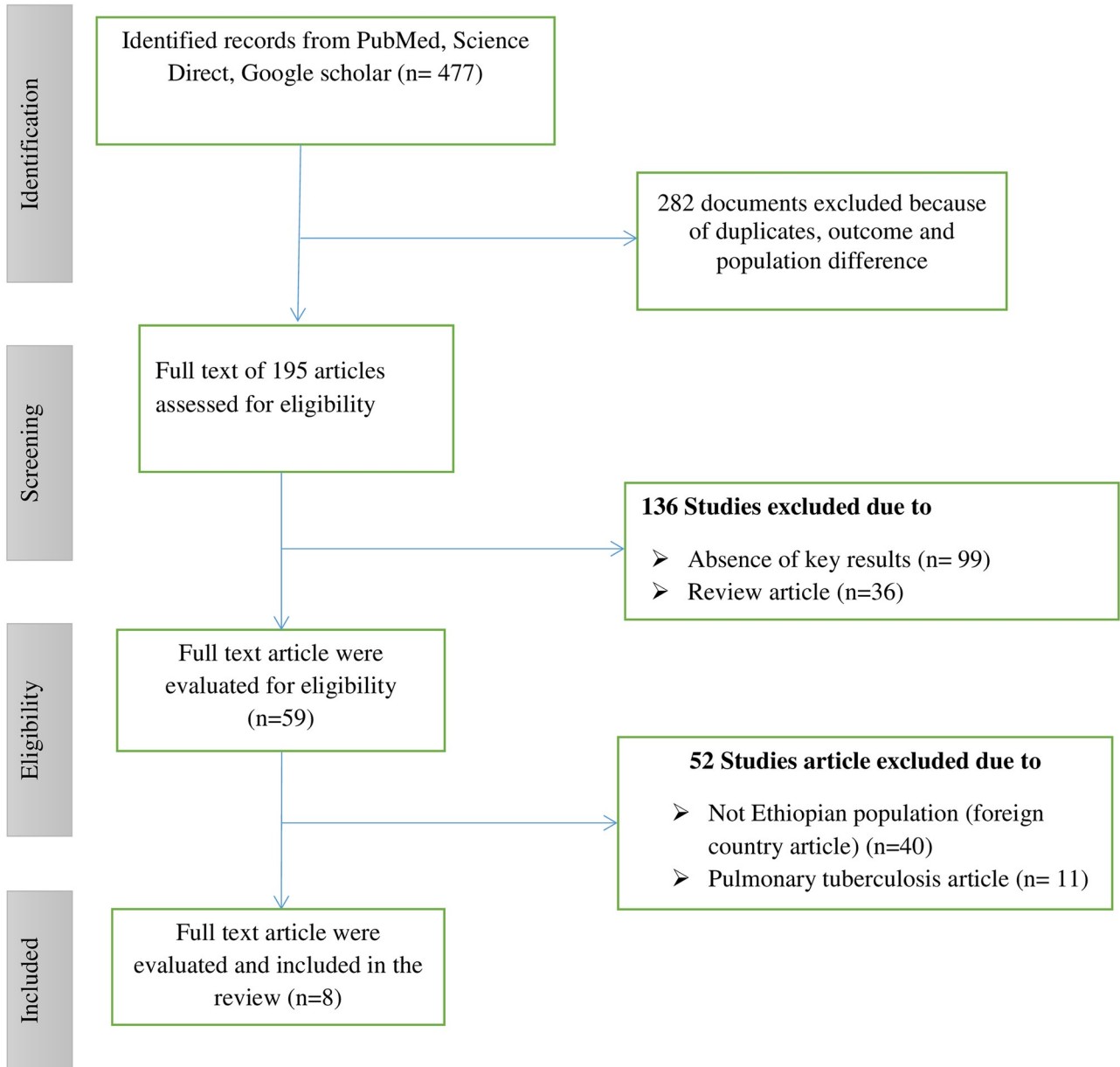

**Fig 1. Flow diagram of the study search, selection and screening literature for the review.**

The most commonly reported risk factor of MDR-TB occurrence was having previous exposure to anti-TB treatment [22]. In contrast, two studies [25, 27] have demonstrated the preventive effect of previous anti-TB exposure. Male sex was also reported as the risk factor of MDR-TB occurrence [22].

## Discussion

In this review, eight cross-sectional studies published between 2014 and 2020 were included. This review estimated the pooled prevalence of first-line anti-tubercular drug resistance and the risk factors of DR-TB occurrence in patients with EPTB in Ethiopia. The pooled prevalence

**Table 1. General characteristics of studies included in the review.**

| First Author, Year [ref.] | Study Design | Study area | Study Setting | Sample size | RIF Resistant | INH Resistant | MDR-TB | DST method |
|---|---|---|---|---|---|---|---|---|
| Biadglegne et al, 2014 [27] | Cross sectional | Bahirdar, Gondar & Dessie | Facility based | 225 | 4 | 8 | 3 | Phenotypic/ MGIT 960 system |
| Mulu et al, 2017 [22] | Cross sectional | Debre Markos | Facility based | 53 | 6 | 0 | 0 | Genotypic (Xpert MTB/RIF) |
| Bekele et al, 2018 [24] | Cross sectional | North Ethiopia | Facility based | 54 | 3 | 0 | 2 | Genotypic/ MTBDR*plus* |
| Tadesse et al, 2015 [23] | Cross sectional | Jimma | Facility based | 92 | 4 | 0 | 0 | Genotypic Xpert MTB/RIF |
| Korma et al, 2015 [28] | Cross sectional | Addis Ababa | Facility based | 59 | 13 | 5 | 0 | Phenotypic/ MGIT 960 system |
| Zewdie et al, 2018 [25] | Cross sectional | Addis Ababa | Facility based | 60 | 5 | 6 | 5 | Genotypic/ MTBDR*plus* |
| Sitotaw et al, 2017 [26] | Cross sectional | Bahirdar | Facility based | 82 | 2 | 3 | 1 | Genotypic/ MTBDR*plus* |
| Diriba et al, 2020 [29] | Cross sectional | Addis Ababa | Facility based | 151 | 14 | 22 | 14 | Phenotypic/ MGIT 960 system |

DST-drug susceptibility test, RIF-rifampicin, INH-Isoniazid, MDR-TB-multidrug resistance tuberculosis.

of INH resistance was 7.0%, while RIF resistance was 6% and MDR-TB was 4%. The two risk factors of DR-TB occurrence were previous TB treatment history and being male sex.

In this review, the pooled prevalence of RIF resistance was 6%. The pooled prevalence of the present review was relatively higher than in the previous primary study reported from Iran, in which the prevalence of RIF resistance TB was 3.1% [30]. Moreover, the results of the current review findings are relatively similar to a review study reported from Saudi Arabia in which the pooled prevalence of RIF-resistant TB is 5.4% [31]. In contrast, the previous studies reported from different countries indicated a higher prevalence of RIF resistance; 21% [32] and 8,6% [33], than in our finding. This substantial difference could be due to drug-resistance detection methods and the variability in the burden of drug resistance across study sites within the country and/or by country. The studies included in the current review employed different

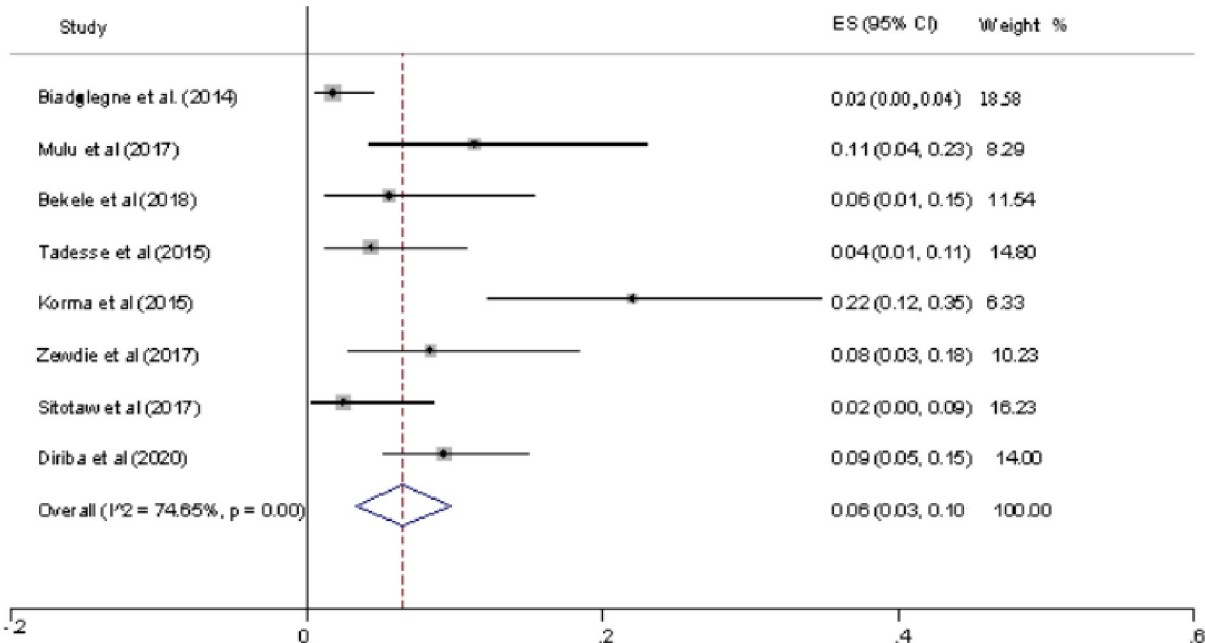

**Fig 2. Forest plot showing the prevalence of RIF among the total sample.**

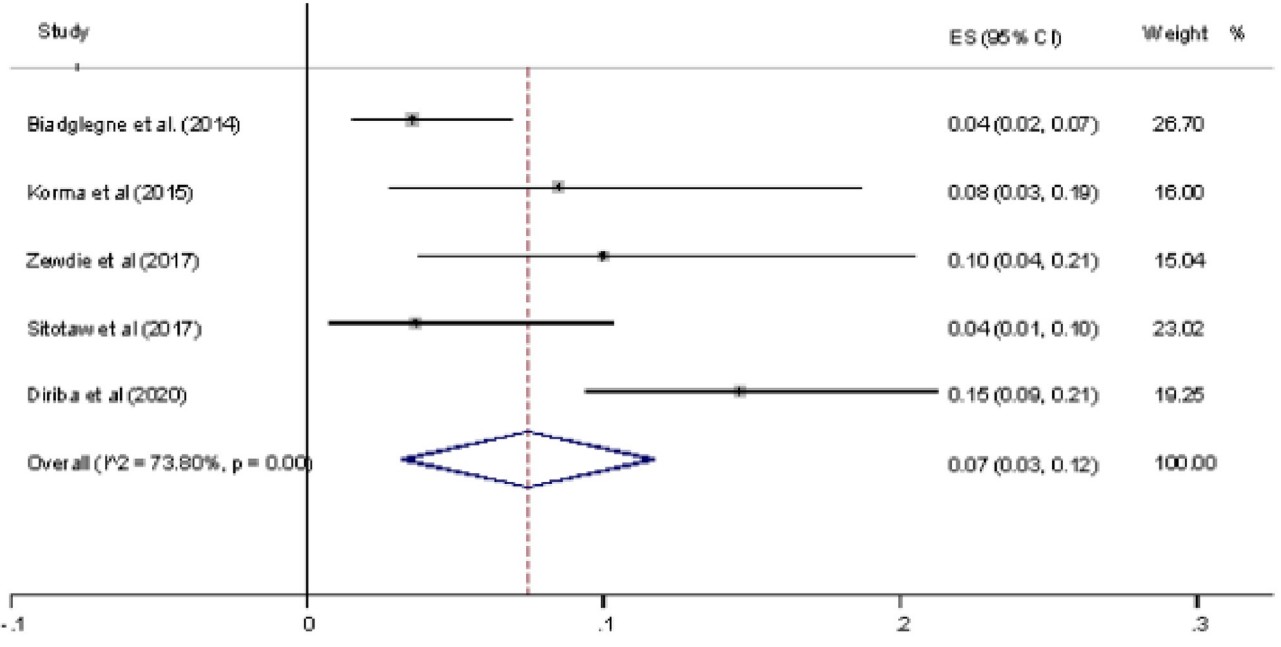

**Fig 3. Forest plot showing the prevalence of INH among the total sample.**

susceptibility testing methods, such as the Xpert MTB/RIF assay, the Genotype MTBDR*plus* assay, and phenotypic methods like the BACTEC system. The previous studies might be used only genotypic or phenotypic methods.

In the present review, the pooled prevalence of INH resistance was 7% in patient with EPTB. Our finding is in agreement with the previous review studies in which the pooled

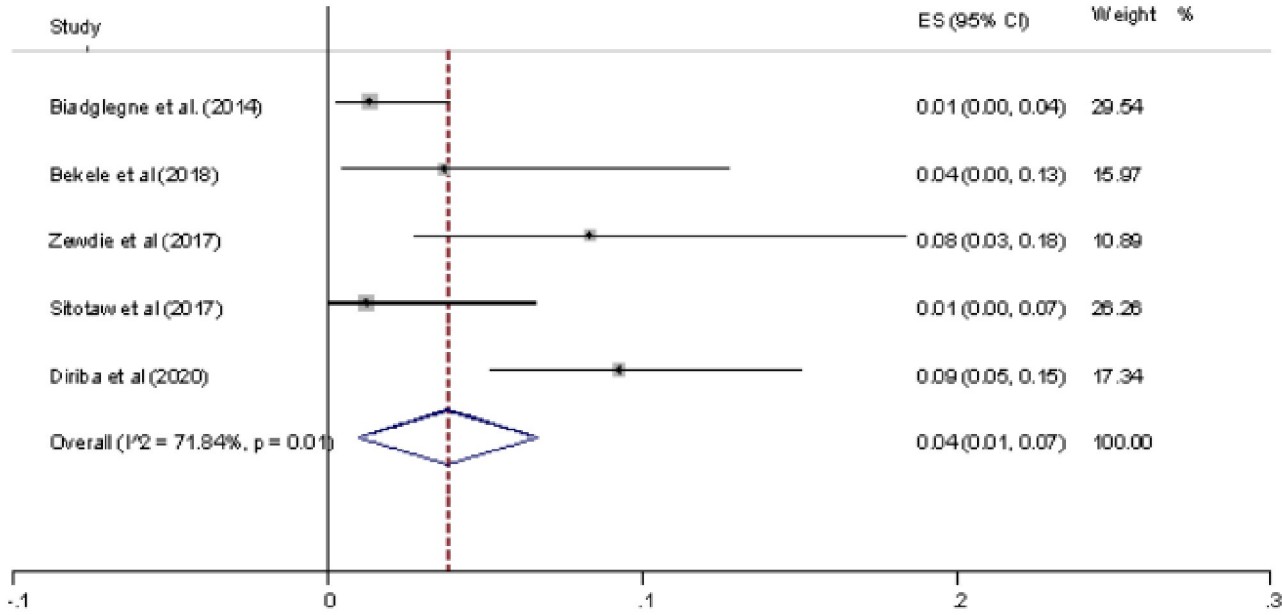

**Fig 4. Forest plot showing the prevalence of MDR-TB among the total sample (EPTB) case (the pooled estimate from the random-effects model).**

prevalence of INH resistance was 8% [15, 34]. However, the prevalence of INH resistance in the current review was lower than the results of a previous review study reported from Iran in which the prevalence of INH resistance was 12.8% in new TB cases and 40.1% in previously treated cases [35]. The reason is that the DR TB burden in Iran is likely to be higher in Ethiopia, and also in terms of HIV prevalence, as in the current review, the pooled prevalence of MDR-TB was 4% among confirmed EPTB cases, which is consistent with previous studies [22–29]. However, the results of our review indicated a lower prevalence of MDR-TB than reports from previous studies in which the prevalence of those previously treated was 18% and slightly higher than the rate among those newly diagnosed of 3.3% [36]. This might be due to study design differences, where the 2020 MDR-TB WHO report's estimated prevalence was for either RIF resistance or both RIF and INH resistance. Moreover, the WHO surveillance report estimated the prevalence of MDR-TB in new TB cases and previously treated cases separately. In our study, we could not estimate the prevalence of any DR-TB and MDR-TB in new TB cases and retreated TB cases separately due to reported variation in the published papers. Furthermore, these data reported low prevalence rates of MDR-TB, with two studies in India and one study in China finding that the prevalence of MDR-TB among the total cases was respectively 19%, 11.6%, 17.2% [6, 37, 38]. This is quite higher when compared to our study. This might be due to variations in the patient group studied. Most MDR-TB cases are due to poor adherence to TB medications, irregular use of drugs, interrupted drug supplies, physician error, and accessibility of drugs without a prescription [39]. In Ethiopia, the low socioeconomic status of the population, high prevalence of infectious diseases, unfavorable treatment outcomes, longer treatment period, and many more complications make MDR-TB a more complex disease than TB [16].

In our study, we reviewed the risk factors of DR-TB in Ethiopia. This finding occurs at a time when there is more information on rates and factors associated with DR-TB, although data on DR-TB is still limited to the EPTB in Ethiopia. A finding that highlights the association of the previous history of anti-TB with MDR-treatment TB infection was high [9, 27]. As indicated in the study, the observed prevalence of MDR-TB was significantly higher than that of newly diagnosed TB cases. Our results confirm prior reviews that having previous treatment is the most influential risk factor for developing DR-TB and MDR-TB. Similarly, a meta-analysis identified a higher number of patients with a previous history of anti-TB treatment than that of newly diagnosed TB cases [9]. Another systematic review analyzed the risk of MDR-TB was 10.23 times higher in previously treated than in never treated cases [40]. This case occurred due to delayed diagnosis, delayed recognition of drug resistance, inappropriate chemotherapy regimens, inadequate or irregular drug supply, and poor agreement by both patients and clinicians have each been reported as a reason for inadequate treatment.

Our study verified the results of prior meta-analyses showing that the male gender was an essential risk factor for DR-TB and MDR-TB [22]. Similarly, a meta-analysis identified the male gender as being associated with acquired drug resistance [40]. Moreover, a systematic review that investigated the risk factors of DR-TB in different countries using primary studies reported a higher risk of drug resistance among men [41, 42]. It was thought that the possible reason behind men having a higher rate of MDR-TB than women might be due to social and health-seeking behavior differences and higher exposure of males to the outer environment, smoking, alcohol abuse, intravenous drug abuse dependency, and imprisonment status, in which more men than women are involved.

There are limitations to our study. First, due to the limited number of published articles in the country in the specified EPTB data sources, the number of articles reviewed was small. Second, we were not able to determine the resistance rates based on categorizing cases of new and retreated EPTB patients. Third, our analysis may not fully represent the frequency of

extrapulmonary DR-TB as the extent of drug resistance has not yet been fully investigated in Ethiopia. However, the results remain important as the increasing level of drug resistance among EPTB patients of the population is alarming.

## Conclusion

In conclusion, our systematic review showed a high proportion of RIF, INH, and MDR-TB among EPTB patients in Ethiopia. The review showed that the prevalence of extrapulmonary DR-TB has continued to become a serious public health problem in Ethiopia. To our knowledge, this finding could help the programmatic management of the disease within the context of the National TB Control program. Clinicians should request drug susceptibility testing for all patients with presumptive EPTB to detect drug resistance. Our findings highlight the need for more studies evaluating drug resistance in EPTB patients.

## Supporting information

**S1 File. Literature search strategy from searched databases.**
(DOCX)

**S2 File. Detailed data of the included studies.**
(XLSX)

**S3 File. PRISMA checklist.**
(DOC)

**S4 File. Newcastle-Ottawa quality assessment scale for cross sectional studies.**
(DOCX)

## Acknowledgments

We acknowledge all the authors of the original studies included in this systematic review and meta-analysis.

## Author Contributions

**Conceptualization:** Getu Diriba, Habteyes Hailu Tola.

**Data curation:** Getu Diriba, Habteyes Hailu Tola, Ayinalem Alemu, Bazezew Yenew, Dinka Fikadu Gamtesa, Abebaw Kebede.

**Formal analysis:** Getu Diriba, Ayinalem Alemu, Bazezew Yenew, Dinka Fikadu Gamtesa.

**Funding acquisition:** Getu Diriba, Abebaw Kebede.

**Investigation:** Getu Diriba.

**Methodology:** Getu Diriba, Habteyes Hailu Tola, Ayinalem Alemu, Abebaw Kebede.

**Software:** Getu Diriba, Dinka Fikadu Gamtesa.

**Validation:** Getu Diriba, Dinka Fikadu Gamtesa.

**Writing – original draft:** Getu Diriba.

**Writing – review & editing:** Getu Diriba, Habteyes Hailu Tola, Ayinalem Alemu, Abebaw Kebede.

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
