## [Decision Letter · Decision Letter 0]

29 Jun 2021

PONE-D-21-12541

Drug resistance and its risk factors among extrapulmonary tuberculosis in Ethiopia: A systematic review and meta-analysis

PLOS ONE

Dear Dr. Getu Diriba,

Thank you for submitting your manuscript to PLOS ONE. After careful consideration, we feel that it has merit but does not fully meet PLOS ONE’s publication criteria as it currently stands. Therefore, we invite you to submit a revised version of the manuscript that addresses the points raised during the review process.

We look forward to receiving your revised manuscript.

Kind regards,

Mohammad Mehdi Feizabadi, phd

Academic Editor

PLOS ONE

Journal Requirements:

2. In your Methods, please state the timeline of your literature search.

Reviewers' comments:

Reviewer's Responses to Questions

**Comments to the Author**

1. Is the manuscript technically sound, and do the data support the conclusions?

Reviewer #1: Yes

Reviewer #2: Yes

2. Has the statistical analysis been performed appropriately and rigorously? 

Reviewer #1: Yes

Reviewer #2: Yes

3. Have the authors made all data underlying the findings in their manuscript fully available?

Reviewer #1: Yes

Reviewer #2: Yes

4. Is the manuscript presented in an intelligible fashion and written in standard English?

Reviewer #1: Yes

Reviewer #2: Yes

5. Review Comments to the Author

Reviewer #1: 1. The date of electronic database searching is not mentioned in method or else, maybe it need updating electronic database searching.

2. On fig 1, out of 195 studies 136 studies were excluded in which 59 were eligible for full text

evaluation, but the authors wrote 60.

3. Lack of consistency of using words through out of the manuscript for example in line 34, 37 ‘extrapulmonary but in line 54 extra-pulmonary

Further proofread and copyediting are required for this manuscript. The paper must be carefully checked by a native English speaker.

Reviewer #2: 1. Literature Searches and Search terms are incomplete. This is suboptimal for publication for systematic review. Please attach search terms that were used in each database as supplement for Data source and search strategies in the manuscript. Please provide details search terms in supplementary documents. Please attach syntax used in each database as supplementary. Authors should also search Embase in their study.

2. When Pubmed is used for the search, MESH terms are always recommended to be included.

3. Evaluation of gray literature is unclear.

4. Please report here the process of search and inclusion/exclusion of the study and the reasons of exclusion in detail. Inclusion and exclusion criteria should be better detailed (e.g., epidemiological study design).

5. The method for selecting studies is not clear and need further explanation.

6. Quality assessments for all included papers should be shown as a supplementary file.

7. Why authors used "Random effects model". More details are needed to explain the statistical plan.

8. Applying egger weighted regression method was stated in statistical methods but was not reported in result.

9. There is substantive heterogeneity in outcomes, which the authors have reported but have not done anything.

The random-effects model is not a good solution for a high source of heterogeneity. The author can conduct a meta-analysis in sub-groups and report the possible sources of heterogeneity.

10. To investigate the publication bias, a funnel plot has been used. Since this plot and other methods of evaluation, the publication bias in this study is based on the value of the effect size and the standard error, and in descriptive studies, there is no effect size. What do

they represent?

6. PLOS authors have the option to publish the peer review history of their article (what does this mean?). If published, this will include your full peer review and any attached files.

Reviewer #1: No

Reviewer #2: **Yes: **Mohammad Javad Nasiri

---

## [Author Response · Author response to Decision Letter 0]

17 Jul 2021

Review Comments to the Author

Reviewer #1: 

Comment #1: The date of electronic database searching is not mentioned in method or else, maybe it needs updating electronic database searching.

Response: Thank you very much for your critical observation and we apologize for not indicating the timeline for the literature search. To address this comment, we have added the timeline of our literature search since we were recorded the time of our search period (Line 105-106).

Comment #2: On fig 1, out of 195 studies 136 studies were excluded in which 59 were eligible for full text evaluation, but the authors wrote 60.

Response: Thank you for your insightful comments and we apologize for the error. We have corrected the errors in fig 1 based on your comments.

Comment #3: Lack of consistency of using words through out of the manuscript for example in line 34, 37 ‘extrapulmonary but in line 54 extra-pulmonary

Further proofread and copyediting are required for this manuscript. The paper must be carefully checked by a native English speaker.

Response: We have used the words consistently across the document and the language usage and grammar related errors have been corrected by native English speaker colleagues.

Reviewer #2:

 Comment #1: Literature Searches and Search terms are incomplete. This is suboptimal for publication for systematic review. Please attach search terms that were used in each database as supplement for Data source and search strategies in the manuscript. Please provide details search terms in supplementary documents. Please attach syntax used in each database as supplementary. Authors should also search Embase in their study.

Response: Thank you for your critical observations and informative comments. We have attached our search strategy for the databases we used during our search and those that support advanced search as supporting file name "S1: File". For example, advanced search for Google scholar cannot be supported and we just used PICO based on the title only. You are absolutely right, EMBASE should be searched. However, since EMBASE is not freely accessed in our setup, we could not search it. We believe that the majority of the articles published from Ethiopia are on MEDLINE/PubMed, which can be accessed through PubMed search. Thus, the articles that would be missed by our search are small. 

Comment #2: When Pubmed is used for the search, MESH terms are always recommended to be included.

Response: Indeed, we apologize for not indicating as we used MESH term during PubMed search. To address this comment, we have added the phrase which indicates that we used MESH term in our search strategy (Line 110).

Comment #3: Evaluation of gray literature is unclear.

Response: We have evaluated the quality of gray literature as any article published as a peer reviewed article. However, we included only one gray literature, which could not seriously affect the quality of critical appraisal. 

Comment #4: Please report here the process of search and inclusion/exclusion of the study and the reasons of exclusion in detail. Inclusion and exclusion criteria should be better detailed (e.g., epidemiological study design).

Response: Thank you very much for your critical observation. We have addressed your comment by elaborating the inclusion and exclusion criteria of the study in detail (line 113-121).

Comment #5: The method for selecting studies is not clear and need further explanation.

Response: The PRISMA flow chart clearly indicates the study selection process of our review. In addition to the PRISMA flow chart, the text we have provided on page # 6; lines 123 – 130 under the subheading study selection clearly indicated our study selection process. To address your comment, we have elaborated on our selection criteria during inclusion and exclusion criteria provision.

Comment #6: Quality assessments for all included papers should be shown as a supplementary file.

Response: Thank you for your insightful comments and suggestions. Based on your comments, we have corrected and attached the quality results of each included study as the supplementary file.

Comment #7: Why authors used "Random effects model". More details are needed to explain the statistical plan.

Response: The random-effects model was used because of the heterogeneity of the true effect sizes of the included studies. Thus, we believe the explanation provided is enough for the readers. Interested body can read reference books that explain the random-effects model. 

Comment #8: Applying egger weighted regression method was stated in statistical methods but was not reported in result. 

Response: We apologize for not reporting the result of publication bias in the previous version of our manuscript. To address this comment, we have added the results of the funnel plot and Egger test to show the presence of publication bias (Line196-199; page # 9)

Comment #9: There is substantive heterogeneity in outcomes, which the authors have reported but have not done anything. The random-effects model is not a good solution for a high source of heterogeneity. The author can conduct a meta-analysis in sub-groups and report the possible sources of heterogeneity.

Response: Yes, you are right to conduct sub-group analysis to assess the reason for heterogeneity. However, since the studies included in this review were few, they cannot allow us to conduct sub-group analysis based on several factors. Moreover, the random-effects model is the recommended model when there is heterogeneity between the true effect sizes, as we have indicated in the statistical analysis part of the manuscript. 

Comment #10: To investigate the publication bias, a funnel plot has been used. Since this plot and other methods of evaluation, the publication bias in this study is based on the value of the effect size and the standard error, and in descriptive studies, there is no effect size. What do

they represent?

Response: The study specific and pooled results are presented on the forest plots (Fig 2 to Fig 4). Thus, reporting standard errors and effect sizes separately are meaningless.

Yours, Sincerely 

Getu Diriba

---

## [Decision Letter · Decision Letter 1]

24 Sep 2021

Drug resistance and its risk factors among extrapulmonary tuberculosis in Ethiopia: A systematic review and meta-analysis

PONE-D-21-12541R1

Dear Dr. Getu Diriba 

We’re pleased to inform you that your manuscript has been judged scientifically suitable for publication and will be formally accepted for publication once it meets all outstanding technical requirements.

Kind regards,

Mohammad Mehdi Feizabadi, PhD

Academic Editor

PLOS ONE

Additional Editor Comments (optional):

Reviewers' comments:

Reviewer's Responses to Questions

**Comments to the Author**

1. If the authors have adequately addressed your comments raised in a previous round of review and you feel that this manuscript is now acceptable for publication, you may indicate that here to bypass the “Comments to the Author” section, enter your conflict of interest statement in the “Confidential to Editor” section, and submit your "Accept" recommendation.

Reviewer #2: (No Response)

2. Is the manuscript technically sound, and do the data support the conclusions?

Reviewer #2: (No Response)

3. Has the statistical analysis been performed appropriately and rigorously? 

Reviewer #2: (No Response)

4. Have the authors made all data underlying the findings in their manuscript fully available?

Reviewer #2: (No Response)

5. Is the manuscript presented in an intelligible fashion and written in standard English?

Reviewer #2: (No Response)

6. Review Comments to the Author

Reviewer #2: (No Response)

7. PLOS authors have the option to publish the peer review history of their article (what does this mean?). If published, this will include your full peer review and any attached files.

Reviewer #2: **Yes: **Mohammad Javad Nasiri

---

## [Editor Report · Acceptance letter]

29 Sep 2021

PONE-D-21-12541R1 

Drug resistance and its risk factors among extrapulmonary tuberculosis in Ethiopia: A systematic review and meta-analysis 

Dear Dr. Diriba:

I'm pleased to inform you that your manuscript has been deemed suitable for publication in PLOS ONE. Congratulations! Your manuscript is now with our production department. 

Kind regards, 

on behalf of

Dr. Mohammad Mehdi Feizabadi 

Academic Editor

PLOS ONE